# OBJECTS IN SEMANTIC TOPOLOGY

**Shuo Yang**[1]   **Peize Sun**[2]   **Yi Jiang**[3]   **Xiaobo Xia**[4]   **Ruiheng Zhang**[5]
**Zehuan Yuan**[3]   **Changhu Wang**[3]   **Ping Luo**[2]   **Min Xu**[1]*
[1]University of Technology Sydney   [2]The University of Hong Kong
[3]ByteDance AI Lab   [4]University of Sydney   [5]Beijing Institute of Technology

## ABSTRACT

A more realistic object detection paradigm, Open-World Object Detection, has arised increasing research interests in the community recently. A qualified open-world object detector can not only identify objects of known categories, but also discover unknown objects, and incrementally learn to categorize them when their annotations progressively arrive. Previous works rely on independent modules to recognize unknown categories and perform incremental learning, respectively. In this paper, we provide a unified perspective: *Semantic Topology*. During the life-long learning of an open-world object detector, all object instances from the same category are assigned to their corresponding pre-defined node in the semantic topology, including the 'unknown' category. This constraint builds up discriminative feature representations and consistent relationships among objects, thus enabling the detector to distinguish unknown objects out of the known categories, as well as making learned features of known objects undistorted when learning new categories incrementally. Extensive experiments demonstrate that semantic topology, either randomly-generated or derived from a well-trained language model, could outperform the current state-of-the-art open-world object detectors by a large margin, *e.g.*, the absolute open-set error (the number of unknown instances that are wrongly labeled as known) is reduced from 7832 to 2546, exhibiting the inherent superiority of semantic topology on open-world object detection.

## 1 INTRODUCTION

Object detection, which aims at localizing and classifying objects in a given scene (Felzenszwalb et al., 2010; Everingham et al., 2010; Lin et al., 2014), is one of the most iconic abilities of biological intelligence. It was introduced to the artificial intelligence field to endow an intelligence agent with the ability of scene understanding. Although significant advances have been made to improve the object detection system in recent years (Girshick et al., 2014; Ren et al., 2015; Cai & Vasconcelos, 2018; Sun et al., 2020b; Redmon et al., 2016; Lin et al., 2017; Tian et al., 2019; Zhou et al., 2019a; Carion et al., 2020; Sun et al., 2020a), a strong assumption that all the objects of interest have been annotated in the training set, *i.e.*, *close-set learning*, is always made but not holds well. All unknown objects are treated as background and ignored by current detectors, making such detectors cannot handle corner cases in many real-world applications such as autonomous driving, where some obstacles are not available during training, but must be detected when intelligent cars are running on the road. However, creating a large-scale dataset that contains annotations for all objects of interest at once is extremely expensive, even impossible.

Superior to current detectors, humans naturally have the ability to discover both known and unknown objects and gradually learn novel concepts motivated by their curiosity. Learning by discovering the unknown is crucial for human intelligence (Livio, 2017; Meacham, 1983), and has been considered as a key step to achieve artificial general intelligence (AGI) (Goertzel, 2014). Recently, a new object detection paradigm, named *Open-World Object Detection*, has been established (Joseph et al., 2021; Miller et al., 2021; 2018c; Liu et al., 2020) to mimic this learning procedure.

A qualified open-world object detector can not only identify known objects, but also discover object instances of unknown categories and gradually learn to recognize them when their annotations

---
*corresponding author

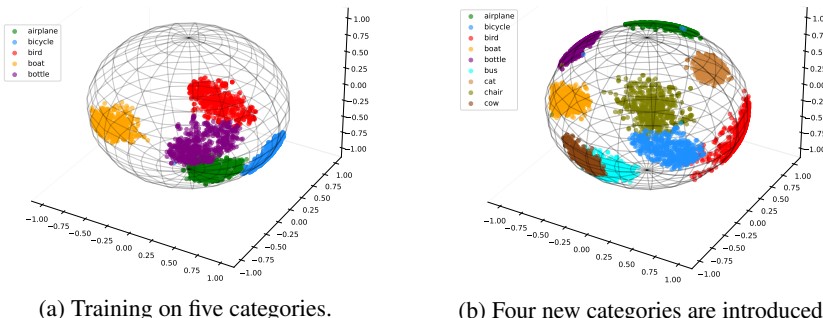

(a) Training on five categories.    (b) Four new categories are introduced.

Figure 1: **t-SNE visualization of object features in ORE.** The location of previously-known (the first five classes) object features are severely *distorted* when learning new categories.

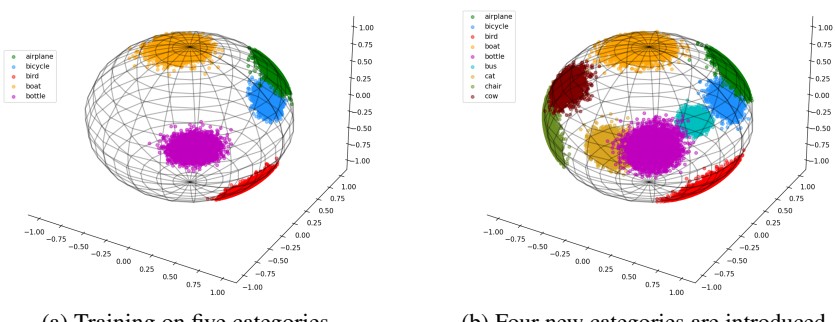

(a) Training on five categories.    (b) Four new categories are introduced.

Figure 2: **t-SNE visualization of object features with our proposed semantic topology.** Both previously-known categories and novel categories are *binded* to their corresponding nodes on the semantic topology. It maintains the previously-known category feature topology when learning novel categories.

progressively arrive. The learning of novel categories is always in an incremental fashion, where the detector cannot access all old data when training on new categories. This *open-world learning* setting is much more realistic but challenging than previous close-set object detection.

The open-world object detection poses two challenges on current detectors, *i.e.*, recognition of unknown categories and incremental learning. On the one hand, previous close-set detectors do not explicitly encourage intra-class compactness (Liu et al., 2016b; Yang et al., 2021b). However, compact feature representation is highly required for unknown object recognition (discovery). If the known categories occupy most of the feature space, the detector will probably classify the unknown object as one of the known categories. On the other hand, the vanilla training strategy of the object detector lacks the mechanism to prevent 'catastrophic forgetting' in incremental learning (Joseph et al., 2021; Shmelkov et al., 2017a; Peng et al., 2020), *i.e.*, previously-known objects features are severely distorted when learning new categories. Consequently, training on novel categories weakens the detector's ability to detect previously-known objects.

Previous works have made efforts to endow the object detector with the capacity of unknown recognition and incremental learning. One of the representative works, ORE (Joseph et al., 2021), designs a clustering loss function to compact the object features, and involves an energy-based out-of-distribution identification approach (Liu et al., 2020) to detect unknown objects. Combining these two independent technologies in a step-wise manner makes ORE a non-end-to-end framework. Such solutions for open-world object detection are far from the optimal status. Additionally, ORE (Joseph et al., 2021) doesn't guarantee a feature space topology consistency, which is crucial for effective new class learning and avoidance of catastrophic forgetting, as shown in Figure 1.

This paper formalizes unknown recognition and incremental learning in a unified perspective and proposes a much simpler but more effective framework for open-world object detection than prior arts. We propose that an open-world object detector is desired to learn a feature space with such characteristics, including (a) *discriminativeness*: the unknown objects and the objects of new categories couldn't be overlapped with any previously-known categories in the feature space, and (b) *consistency*: the feature topology of previously-known categories couldn't be distorted severely when

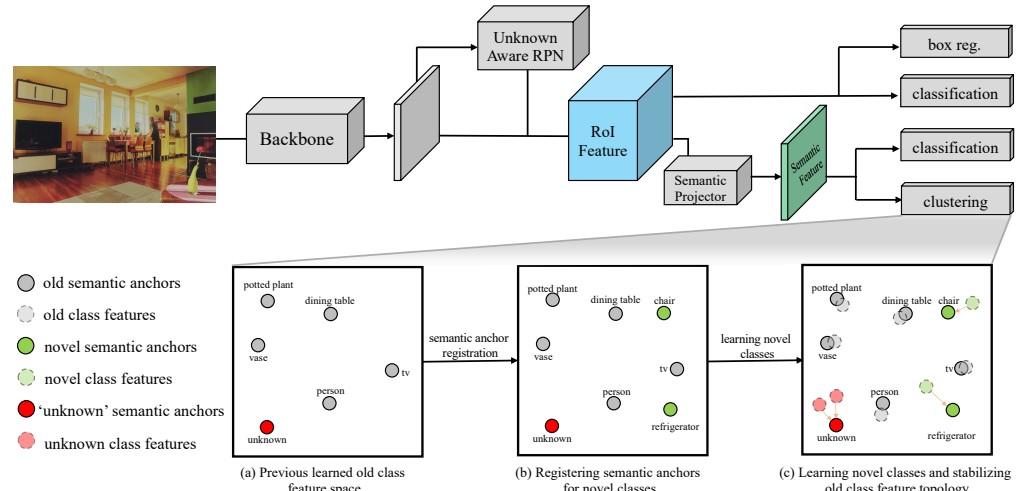

Figure 3: **Illustration of two novel classes, *e.g.*, chair, and refrigerator, are introduced to the open-world object detector at a specific time point during the training procedure**. Each node in the semantic topology, termed as a semantic anchor, is pre-defined by a randomly-generated vector or derived from a well-trained language model before starting the training procedure. When the detector learns novel categories, the corresponding semantic anchors are registered to the semantic topology firstly, then object features of the same category are constrained to close to its semantic anchor. At the inference stage, the RoI feature classifier and the semantic feature classifier are ensembled to make predictions.

learning new categories. Our key idea is to pre-define a unique and fixed centroid in feature space for each category, including the 'unknown' category, and to push object instances close to their belonging centroids during the life-long learning of an open-world object detector. The pre-defined centroids are named as 'Semantic Anchors', and all semantic anchors constitute the structure of 'Semantic Topology'. As shown in Figure 2, all features are binded to their corresponding semantic anchors to satisfy the *discriminativeness*, previously-known objects feature topology is maintained when incrementally learning new categories to satisfy the *consistency*.

We introduce an off-the-shelf pre-trained language model to set up the semantic topology. The semantic anchor for each category is derived from the language model by embedding the corresponding category name. By incrementally registering new semantic anchors when new classes are involved, the semantic topology gradually grows. In the experiments, we show that by combining a current detector with our proposed semantic anchor head, a huge improvement in open-world object detection performance across the board is achieved. In addition to the consistent mAP improvement over the whole life of the detector, the ability of unknown recognition of our proposed method outperforms the current state-of-the-art methods by a large margin, *e.g.*, the absolute open-set error is reduced by 2/3, from 7832 to 2546.

More importantly, we conduct a comparison experiment by randomly generating semantic anchors instead of leveraging language models. Although randomly-generated anchors do not provide semantic priors, their performance still surpasses state-of-the-art methods. This strongly proves that the *topology consistency* is the most key characteristic for open-world learning while introducing semantic relationships can further boost the performance.

## 2 RELATED WORKS

**Object detection**

Modern object detection frameworks (Ren et al., 2015; Redmon et al., 2016; Liu et al., 2016a; Lin et al., 2017; Zhang et al., 2019a; Redmon et al., 2016; Redmon & Farhadi, 2018; Girshick, 2015; Duan et al., 2019; Tian et al., 2019; Tan et al., 2020; Zhou et al., 2019b; Jiang et al., 2018; Zhang et al., 2019b; Carion et al., 2020; Sun et al., 2020b) take advantage of the high-capacity representation in deep neural networks to localize and classify the target class object in given images and videos. These well-established detectors can achieve excellent performance in the close-set dataset such as PascalVOC (Everingham et al., 2010), MSCOCO (Lin et al., 2014). However, the detectors can

not handle open-world object detection, which is more common in the real world. To this end, ORE (Joseph et al., 2021) raises and formalizes the open-world object detection problem.

**Class incremental learning** Class incremental learning aims to learn a classifier incrementally to recognize all encountered classes met so far, including previously-known classes and novel classes. Knowledge distillation is commonly adopted to mitigate forgetting old classes, which stores some old class exemplars to fine-tune the model or compute the distillation loss. iCaRL (Rebuffi et al., 2017) maintains an 'episodic memory' of the exemplars and incrementally adds novel class examples into the memory. Then the nearest neighbor classifier can be obtained incrementally. LwF (Li & Hoiem, 2017) proposes a modified cross-entropy loss to preserve the knowledge in the previous task. BiC (Wu et al., 2019) points out the data imbalance between old classes and new classes causes the network's prediction biased towards new classes. However, the existing class incremental methods cannot handle the open-world problem, where the classifier should identify those 'unknown' classes and incrementally learns to recognize them, but existing methods would recognize unknown objects as background.

**Open-set Learning** In open-set learning, the knowledge contained in the training set is incomplete, *i.e.*, the examples encountered during inference may belong to a category that does not appear in the training set. OpenMax (Bendale & Boult, 2016) uses a Weibull distribution to identify unknown instances in the feature space of deep networks. OLTR (Liu et al., 2019) proposes to tackle the open-set recognition problem in a data imbalance setting by deep metric learning. Besides the open-set classification, Dhamija *et al.* (Dhamija et al., 2020b) found that the object detectors exhibit a high error rate that misclassifies unknown classes to known classes with high confidence. To solve this problem, many works (Miller et al., 2021; 2018a; Dhamija et al., 2020b) aims to measure the uncertainty of the detectors' outputs to reject open-set error. Miller *et al.* (Miller et al., 2018a) uses Monte Carlo Dropout sampling to estimate the uncertainty in an SSD detector. After that, Miller *et al.* proposes to model a Gaussian Mixture distribution for each class in the detector's feature space to reject unknown classes. ORE (Joseph et al., 2021) uses an energy-based out-of-distribution recognition method (Liu et al., 2020) to distinguish the known and unknown. However, such method needs to compute the energy score distribution among all instances, including known and unknown, during evaluation, making ORE a non-end-to-end method. We formulate the open-set recognition problem and the class incremental learning into a unified framework. Different from (Joseph et al., 2021), our method doesn't access any unknown instances but achieves much superior performance.

**Zero-shot Learning** Aligning image and text into a common feature space has always been an active research topic (Frome et al., 2013; Joulin et al., 2016; Li et al., 2017; Desai & Johnson, 2021; Yang et al., 2019; 2021a), especially in zero-shot learning (Xian et al., 2017; Bansal et al., 2018; Zareian et al., 2021; Gu et al., 2021). Many researches in zero-shot learning leverage the information in language models to assist zero-shot image classification (Xian et al., 2017) or zero-shot object detection (Bansal et al., 2018; Zareian et al., 2021; Gu et al., 2021). However, this paper tackles a different problem and has a different usage of language models. We identify that *a consistent feature manifold topology plays an essential role in open-world object detection*, and the language model is used for generating growth-able and consistent semantic topology to constrain the feature space learning of an open-world detection. Also, this paper is the first one to incorporate a language model, *e.g.*, CLIP, to assist open-world object detection, which results in a simple training framework and strong empirical performance.

## 3 METHODOLOGY

In this section, we introduce the *Open World Object Detection* problem definition in Section 3.1, the method overview in Section 3.2, and the details of the proposed method in Section 3.3 and Section 3.4.

### 3.1 PROBLEM DEFINITION

An open world object detector should detect all the previously seen object classes, and can also identify if a testing instance known or unknown (belongs to previously seen classes or not). If unknown, the detector should gradually learn the unknown classes when their annotations progressively arrive without retraining from scratch. Formally, at each time point $t$, we assume there exists a set of known object classes $\mathcal{C}_{kn}^t = \{l_1, l_2, \ldots, l_C\}$ and a set of unknown object classes $\mathcal{C}_{unk}^t = \{l_{C+1}, l_{C+2}, \ldots\}$. The detector $\mathcal{D}_t$ at the time point $t$ has only trained on classes in $\mathcal{C}_{kn}^t$ while may encounter all classes

including $\mathcal{C}_{kn}^t$ and $\mathcal{C}_{ukn}^t$ during evaluation. Besides correctly classifying an object instance from the known classes $\mathcal{C}_{kn}^t$, the detector $\mathcal{D}_t$ should also label all instances from the unknown class set $\mathcal{C}_{ukn}^t$ as unknown. At the time point $t+1$, the unknown instances will be forwarded to a human user who can select $n$ novel classes of interest to annotate and return them back to the model. The detector $\mathcal{D}_t$ should incrementally learn these $n$ novel classes and updates itself to $\mathcal{D}_{t+1}$ without retraining from scratch on the whole dataset. The known class set $\mathcal{C}_{kn}^{t+1}$ in the time point $t+1$ is also updated to $\mathcal{C}_{kn}^{t+1} = \mathcal{C}_{kn}^t + \{l_{C+1}, \ldots, l_{C+n}\}$. The data instances in the known classes $\mathcal{C}_{kn}$ are assumed to be labeled in the form of $\{x, y\}$, where $x$ indicates the image and $y$ indicates the annotations including class label $l$ and object coordinates, i.e., $y = [l, x, y, w, h]$ where $x, y, w, h$ denote the bounding box center coordinates, width and height respectively.

## 3.2 METHOD OVERVIEW

A Region Proposal Network (RPN) that can identify unknown and a *discriminative* and *consistent* feature space are two critical components for an open world detector. Here, we adopt the Unknown-Aware RPN proposed in (Joseph et al., 2021) and propose to constrain the detector's feature space topology with a pre-defined *Semantic Topology*. Specifically, we create a unique and fixed centroid for each category in the feature space, named *semantic anchor*. The semantic anchors for all classes are derived by feeding forward their class names into a pre-trained language model. Our key idea is to manipulate the detector's feature space to be consistent with the semantic topology constituted by semantic anchors during the whole life of the detector. Due to the feature dimension discrepancy, a fully connected layer (semantic projector) is used to align the dimension between RoI features and semantic anchors. At the training stage, the semantic features outputted by the semantic projector are forced to cluster around their corresponding semantic anchors by a designed 'SA (semantic anchor) Head'. When incremental learning, the SA Head gradually registers new semantic anchors for novel classes and continually pulls close the novel class features and their semantic anchors. To mitigate 'catastrophic forgetting' caused by old class feature distortion, the SA Head also minimizes the distance between some stored old class exemplars and their semantic anchors when learning novel classes. To better leverage the well-constructed feature space, we attach an additional classification layer to classify the semantic features. Figure 3 shows the training pipeline of the proposed open-world object detector. At inference, we multiply the class posterior probabilities produced by the two classification heads and get the final prediction.

## 3.3 UNKNOWN-AWARE RPN

Open-world object detectors are required to separate potential unknown objects from the background. Therefore, we need some specific designs for the Region Proposal Network (RPN). In this paper, we adopt the unknown-ware RPN proposed in (Joseph et al., 2021) which selects the top-k background region proposals, sorted by its objectness scores, as unknown objects. The unknown-aware RPN relies on the fact that Region Proposal Network is class agnostic. Given an input image, the RPN generates bounding box predictions for foreground and background instances, along with the corresponding objectness scores. The unknown-aware RPN labels those proposals with a high objectness score but do not overlap with any ground-truth object as a potential unknown object.

## 3.4 SEMANTIC TOPOLOGY

We propose to pre-define a *semantic topology* for detectors' feature space rather than learn from data. The semantic topology is constituted by *semantic anchors*. Each semantic anchor is a pre-defined feature centroid for an object class. The semantic anchors can be generated by embedding corresponding class names using a pre-trained language model. The semantic topology can dynamically grow as novel classes are introduced to the open-world detector.

### 3.4.1 SEMANTIC ANCHOR REGISTRATION

We generate semantic anchors for all classes by feeding the class names into an off-the-shelf pre-trained language model. Denote $l_i \in \mathcal{C}_{kn}^t$ as class *name* of the $i$-th known class at time $t$ and $\mathcal{M}$ as an off-the-shelf pre-trained language model. The semantic anchor for class $l_i$ is defined as $\mathcal{A}_i = \mathcal{M}(l_i)$, where $\mathcal{A}_i \in \mathbb{R}^n$, the dimension $n$ depends on the pre-trained language model. The semantic anchor registration is performed repeatedly as long as known class set update $\mathcal{C}_{kn}^t \to \mathcal{C}_{kn}^{t+1}$ when novel classes are introduced at time $t+1$. Note we also register a semantic anchor for all instances labeled

as `unknown` by the unknown-aware RPN to better distinguish the unknown instances from the known instances. Follows the same strategy, the semantic anchor for class `unknown` is also generated by the word embedding of `unknown` text.

### 3.4.2 OBJECTIVE FUNCTION

The RoI features $f \in \mathbb{R}^d$ are feature vectors generated by an intermediate layer of the object detector, which are used for category classification and bounding box regression. We manipulate the RoI features $f$ to construct the detector's feature manifold topology. Denote $f_i$ as an RoI feature of the $i$-th known class, we first align the dimension of $f_i$ as the dimension of its corresponding semantic anchor $\mathcal{A}_i$, using a fully connected layer with $d \times n$ dimensional weights. The corresponding semantic feature is denoted as $\hat{f}_i \in \mathbb{R}^n$. We constrain the detector's feature manifold topology by clustering the semantic features around their corresponding semantic anchors, the learning objective is formalized as

$$\mathcal{L}_{sa} = \|\hat{f}_i - \mathcal{A}_i\|. \tag{1}$$

Minimizing this loss would ensure the desired feature space topology. To better leverage the constructed feature space, we use an additional classification head to classify the semantic features $\hat{f}_i$, with the same label space as RoI classification head. The total training objective is the combination of semantic anchor loss $\mathcal{L}_{sa}$, semantic feature classification loss $\mathcal{L}_{cls_{se}}$, RoI feature classification loss and bounding box regression loss:

$$\mathcal{L}_{total} = \mathcal{L}_{sa} + \mathcal{L}_{cls_{se}} + \mathcal{L}_{cls_{roi}} + \mathcal{L}_{reg}. \tag{2}$$

At the inference stage, the classification results are computed by multiplying the two class posterior probability vectors predicted by the RoI feature classification head and the semantic feature classification head.

### 3.4.3 TOPOLOGY STABILIZATION

To store the detection ability on old classes, a balanced set of exemplars are stored and used to fine-tune the model after each incremental learning session as in the previous open-world object detection approach (Joseph et al., 2021). However, we argue that finetuning the detector ignoring the old knowledge topology as in (Joseph et al., 2021) still suffers from the severe 'catastrophic forgetting' problem. Benefitting from pre-defining the feature space topology, our method can guarantee a consistent feature space during the finetuning stage. The stored old class and new class instances are still forced to cluster around their pre-defined centroids to guarantee the feature space topology unchanged.

## 4 EXPERIMENTS

We introduce the evaluation protocol, including datasets and evaluation metrics, implementation details, and experimental results in this section.

### 4.1 DATASETS

Following (Joseph et al., 2021), the open-world detector is evaluated on all 80 object classes from Pascal VOC (Everingham et al., 2010) (20 classes) and MS-COCO (Lin et al., 2014) (20+60 classes). All categories are grouped into a set of tasks $\mathcal{T} = \{T_1, \dots, T_t, \dots\}$, where all categories of $T_t$ will be introduced to the detector at a time point $t$. At the time point $t$, all categories from $\{T_\tau | \tau <= t\}$ will be treated as known and $\{T_\tau | \tau > t\}$ would be treated as unknown. As in (Joseph et al., 2021), $T_1$ consists of all VOC classes, and the remaining 60 classes from MS-COCO are grouped into three successive tasks with semantic drifts. The open-world object detector is trained on the training set of all classes from Pascal VOC and MS-COCO, and evaluated on the Pascal VOC test split and MS-COCO val split. The validation set consists of 1k images from the training data of each task.

### 4.2 EVALUATION METRICS

We introduce three metrics to evaluate the detection performance of an open-world object detector on known classes and unknown classes at each time point:

| Task IDs (→) | Task 1 | | | Task 2 | | | | | Task 3 | | | | | Task 4 | | |
|---|---|---|---|---|---|---|---|---|---|---|---|---|---|---|---|---|
| | WI | A-OSE | mAP (↑) | WI | A-OSE | mAP (↑) | | | WI | A-OSE | mAP (↑) | | | mAP (↑) | | |
| | (↓) | (↓) | Current known | (↓) | (↓) | Previously known | Current known | Both | (↓) | (↓) | Previously known | Current known | Both | Previously known | Current known | Both |
| Faster R-CNN | 0.06461 | 13286 | 55.95 | 0.0492 | 9881 | 5.29 | 25.36 | 15.32 | 0.0231 | 9294 | 6.09 | 13.53 | 8.570 | 1.98 | 13.95 | 4.97 |
| Faster R-CNN + Finetuning | 0.06461 | 13286 | 55.95 | 0.0523 | 11913 | 51.07 | 23.84 | 37.46 | 0.0288 | 9622 | 35.39 | 11.03 | 27.24 | 29.06 | 12.23 | 24.85 |
| ORE | 0.0477 | 7995 | 56.02 | 0.0297 | 7832 | 52.19 | 25.03 | 38.61 | 0.0218 | 6900 | 37.23 | 12.02 | 28.82 | 29.53 | 13.09 | 25.42 |
| Ours | **0.0417** | **4889** | **56.20** | **0.0213** | **2546** | 53.39 | 26.49 | **39.94** | **0.0146** | **2120** | 38.04 | 12.81 | **29.63** | 30.11 | 13.31 | **25.91** |

Table 1: **Comparisons of different methods on Open World Object Detection.** Wilderness Impact (WI) and Absolute Open Set Error (A-OSE) are both the less the better, which measure the ability on unknown identification. Our proposed method achieves consistent performance improvement during the whole life of the detector, surprisingly achieves much lower A-OSE compared with baselines and previous state-of-the-art methods.

- Mean Average Precision (mAP) (Everingham et al., 2010; Lin et al., 2014). Following previous works (Joseph et al., 2021), we use mAP at IoU threshold of 0.5 to measure the performance on known classes. Since the object classes are continually involved, mAP is calculated on previously-known classes and currently-known classes.

- Absolute Open-Set Error (A-OSE) (Joseph et al., 2021; Miller et al., 2018b). A-OSE counts the absolute number of unknown instances that are wrongly classified as any of the known classes. A-OSE evaluates the detector's ability to avoid misclassifying unknown instances as one of the known classes.

- Wilderness Impact (WI) (Joseph et al., 2021; Dhamija et al., 2020a). The definition is WI $= \frac{P_{\mathcal{K}}}{P_{\mathcal{K} \cup \mathcal{U}}} - 1$, where $P_{\mathcal{K}}$ is the precision of the detector when evaluated on known classes and $P_{\mathcal{K} \cup \mathcal{U}}$ refers to the precision when evaluated on all classes including known and unknown. The recall level $R$ is set to be 0.8. WI evaluates the detector's ability to successfully detect unknown objects and classify them as 'unknown' class.

### 4.3 IMPLEMENTATION DETAILS

We extend the traditional Faster R-CNN (Ren et al., 2015) object detector with a ResNet-50 (He et al., 2016) to be an open-world object detector. The RoI feature is extracted from the last residual block in the RoI head, which has a 2048 dimension. The 2048-dim RoI feature is used for computing traditional bounding box regression loss and classification loss as the same as in many previous works (Joseph et al., 2021; Ren et al., 2015). The semantic projector is a fully-connected layer to align the dimension of RoI features with the semantic anchors. The dimension of semantic anchors depends on the choice of pre-trained language model, e.g., the semantic anchor is 512-dim when using CLIP text encoder (Radford et al., 2021b) to embed categories names. The semantic feature outputted by the semantic projector is used for calculating semantic anchor loss $\mathcal{L}_{sa}$ and semantic feature classification $\mathcal{L}_{cls_{se}}$ loss. The $\mathcal{L}_{sa}$ and the $\mathcal{L}_{cls_{se}}$ are added to the standard regression loss and RoI classification loss to jointly optimize the detector. For the topology stabilization, we store 100 instances per class as the same as in (Joseph et al., 2021). To enable the classifier to handle a variable number of classes at different time points, we assume there exists a maximum number of classes to be added and set the classification logits of unseen classes to a large negative value to make their contribution to softmax output negligible, following (Joseph & Balasubramanian, 2020; Rajasegaran et al., 2020; Lopez-Paz & Ranzato, 2017; Chaudhry et al., 2018).

To make a fair comparison, we report experimental results obtained by re-running the ORE (Joseph et al., 2021) official code[1]. All the hyper-parameters and optimizers are also controlled to be exactly the same for all methods.

### 4.4 OPEN-WORLD OBJECT DETECTION

Table 1 shows the open-world object detection performance of our proposed method and other baselines. At task 1, all methods are trained over all 20 Pascal-VOC classes. Twenty novel object classes are introduced gradually at the following task. WI and A-OSE are used to quantify how unknown instances are confused with known class objects after training on each task. WI and A-OSE are both the less the better. ORE (Joseph et al., 2021) used an extra clustering loss to separate class features, and an energy-based unknown identifier that learns shifted Weibull distributions from

---

[1]Official repository: https://github.com/JosephKJ/OWOD

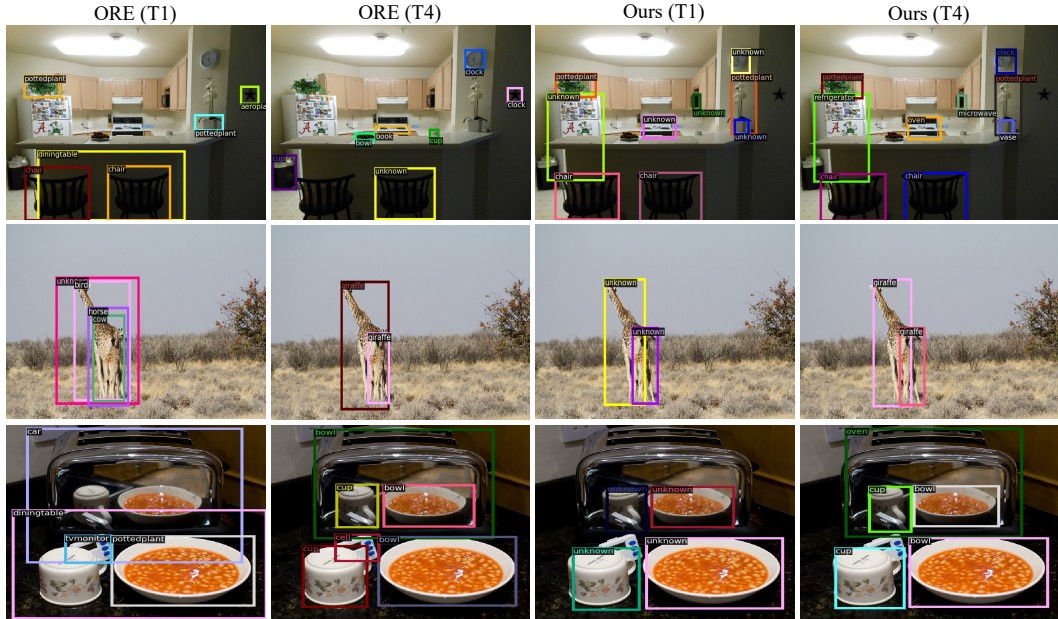

Figure 4: **Visualization of ORE and our proposed method after training on Task 1 (T1) and Task 4 (T4).** At task 1, ORE frequently misclassify unknown object as one of the known object classes. After training on task 4, ORE successfully detects novel classes but forgets objects learned on task 1. By explicitly introducing semantic prior into the detector to constrain the feature space topology, our proposed method performs favorably in open-world object detection.

all class data (including known and unknown) in the validation set to reject open-set error. Our method doesn't access any unknown data instances from the validation set and achieves much superior performance on WI and A-OSE than Faster R-CNN and ORE. In task 3, our proposed method achieves 1/3 A-OSE compared to ORE (2120 v.s. 6900) and 1/4 A-OSE compared to Faster R-CNN (2120 v.s. 9622). Our method also reduces the Wilderness Impact (WI) by a large margin than baselines. This indicates our method has a much better ability on unknown detection. Furthermore, our method also achieves consistent mAP improvement over the whole life of the detector. At incremental learning session (task 2,3,4), the performance of Faster R-CNN on old classes deteriorate severely (55.35 → 6.09). Faster R-CNN + Finetuning, ORE, and our method store some old class instances and fine-tune on them after learning new classes to restore the old class performance. However, the Faster R-CNN + Finetuning and ORE don't guarantee the feature manifold topology consistent. The superior performance on old classes also proves that guaranteeing the feature manifold topology is critical.

## 4.5 INCREMENTAL OBJECT DETECTION

To verify the effectiveness of our proposed method, we also conduct experiments on class incremental object detection (iOD), where the detectors only need to continually recognize novel object classes without unknown identification. We use the standard iOD evaluation protocol as (Joseph et al., 2021; Shmelkov et al., 2017b) to evaluate all methods, where group of classes (10, 5, and the last class) from PASCAL VOC 2007 (Everingham et al., 2010) are incrementally learned by a detector trained on the remaining set of classes. Benefits from the well-constructed feature space using semantic anchors, our proposed method performs favorably well on the incremental object detection (iOD) task against several state-of-the-art 2. Notably, our method exhibits strong ability to incrementally learn new classes while not forget old classes. This is because our detector's feature space is constrained to be incrementally growth-able by assigning each object class a unique feature space location.

## 4.6 ABLATION STUDY

**The effect of language model.** We derive semantic topology from pre-trained language models. To explore the effect of the choice of semantic topology, we conduct ablation studies in this section. To be specific, we generate semantic anchors using two pre-trained language models, *i.e.*, CLIP-text (Radford et al., 2021a) and BERT (Devlin et al., 2019). The dimensions of semantic anchor generated by CLIP-text and BERT are 512 and 768, respectively. The comparison results are shown in

| Task Settings (→) | 10 + 10 | | | 15 + 5 | | | 19 + 1 | | |
|---|---|---|---|---|---|---|---|---|---|
| | mAP (↑) | | | mAP (↑) | | | mAP (↑) | | |
| | old classes | new classes | both | old classes | new classes | both | old classes | new classes | both |
| Joint train | 70.80 | 70.20 | 70.50 | 72.10 | 65.72 | 70.51 | 70.52 | 70.3 | 70.51 |
| ILOD (Shmelkov et al., 2017a) | 63.16 | 63.14 | 63.15 | 68.34 | 58.44 | 65.87 | 68.54 | 62.7 | 68.25 |
| ILOD + Faster R-CNN | 67.33 | 54.93 | 61.14 | 69.24 | 57.56 | 66.35 | 67.81 | 65.1 | 67.72 |
| Faster ILOD (Peng et al., 2020) | 69.76 | 54.47 | 62.16 | 71.56 | 56.94 | 67.94 | 68.91 | 61.1 | 68.56 |
| ORE (Joseph et al., 2021) | 58.37 | 68.70 | 63.53 | 71.44 | 58.33 | 68.17 | 68.95 | 60.1 | 68.50 |
| Ours | 60.03 | 69.88 | **64.96** | 73.01 | 60.69 | **69.93** | 70.22 | 62.30 | **69.82** |

Table 2: **Comparison of different methods on class incremental object detection.** In the three task settings, 10, 5, and the last class from the Pascal VOC 2007 (Everingham et al., 2010) dataset are introduced to a detector trained on 10, 15, and 19 classes respectively.

| Task IDs (→) | Task 1 | | | Task 2 | | | | | Task 3 | | | | | Task 4 | | |
|---|---|---|---|---|---|---|---|---|---|---|---|---|---|---|---|---|
| | WI | A-OSE | mAP (↑) | WI | A-OSE | mAP (↑) | | | WI | A-OSE | mAP (↑) | | | mAP (↑) | | |
| | (↓) | (↓) | Current known | (↓) | (↓) | Previously known | Current known | Both | (↓) | (↓) | Previously known | Current known | Both | Previously known | Current known | Both |
| ORE | 0.0477 | 7995 | 56.02 | 0.0297 | 7832 | 52.19 | 25.03 | 38.61 | 0.0218 | 6900 | 37.23 | 12.02 | 28.82 | 29.53 | 13.09 | 25.42 |
| Random | 0.0433 | 5331 | 56.13 | 0.0246 | 2779 | 52.56 | 25.86 | 39.21 | 0.0183 | 3742 | 37.69 | 12.33 | 29.23 | 29.31 | 12.98 | 25.22 |
| BERT | 0.0421 | 4903 | **56.20** | 0.0222 | 2772 | 53.17 | 25.84 | 39.51 | 0.0153 | 2248 | 37.92 | 12.73 | 29.52 | 30.07 | 13.29 | 25.87 |
| CLIP | **0.0417** | **4889** | **56.20** | **0.0213** | **2546** | 53.39 | 26.49 | **39.94** | **0.0146** | **2120** | 38.04 | 12.81 | **29.63** | 30.11 | 13.31 | **25.91** |

Table 3: **Ablation on semantic anchor generation.** Semantic anchors derived from CLIP-text and BERT obtain the similar performance, both outperforming ORE. Surprisingly, semantic anchors from random vectors also achieve better result than ORE.

| Task IDs (→) | Task 1 | | | Task 2 | | | | | Task 3 | | | | | Task 4 | | |
|---|---|---|---|---|---|---|---|---|---|---|---|---|---|---|---|---|
| | WI | A-OSE | mAP (↑) | WI | A-OSE | mAP (↑) | | | WI | A-OSE | mAP (↑) | | | mAP (↑) | | |
| | (↓) | (↓) | Current known | (↓) | (↓) | Previously known | Current known | Both | (↓) | (↓) | Previously known | Current known | Both | Previously known | Current known | Both |
| w/o $\mathcal{L}_{sa}$ | 0.0641 | 13097 | 55.98 | 0.0317 | 12564 | 51.03 | 23.97 | 37.50 | 0.0269 | 9598 | 35.21 | 11.13 | 27.18 | 28.87 | 12.17 | 24.69 |
| w/o unknown anchor | 0.0476 | 6032 | 56.19 | 0.0270 | 3354 | 52.36 | 25.26 | 38.18 | 0.0193 | 4692 | 37.79 | 12.63 | 29.40 | 29.42 | 13.17 | 25.35 |
| w/o $\mathcal{L}_{cls_{se}}$ | 0.0479 | 12961 | 55.83 | 0.0290 | 11297 | 52.37 | 26.43 | 39.40 | 0.0237 | 8713 | 35.46 | 11.33 | 27.41 | 28.91 | 12.34 | 24.76 |
| w/o $\mathcal{L}_{cls_{roi}}$ | 0.0428 | 5301 | 56.13 | 0.0233 | 2896 | 53.40 | 26.45 | 39.92 | 0.0162 | 2694 | 38.01 | 12.77 | 29.59 | 29.64 | 13.28 | 25.55 |
| our full model | **0.0417** | **4889** | **56.20** | **0.0213** | **2546** | 53.39 | 26.49 | **39.94** | **0.0146** | **2120** | 38.04 | 12.81 | **29.63** | 30.11 | 13.31 | **25.91** |

Table 4: **Ablation on `unknown` anchor and loss components.**

Table 3. These two language models obtain similar results, which both outperform ORE, the previous state-of-the-art method in open-world object detection. To further explore the importance of semantic priors, we also generate semantic anchors by randomly and uniformly sampling, which means we assume the total number of classes is known which is not applicable in practice. Surprisingly, our experiment shows that semantic anchors as random vectors still surpass state-of-the-art methods. This demonstrates that a consistent topology, even without semantic priors, could also largely benefit the task of open-world object detection, which strongly proves that the *discriminative* and the *consistent* are two key characteristics for open-world learning. However, the randomly assigned class centers would hinder the learning ability of networks. Instead, a pretrained language model can perfectly overcome these problem.

**The effect of semantic anchor loss.** In this section, we explore the importance of `unknown` class anchor, semantic anchor clustering loss $\mathcal{L}_{sa}$, semantic anchor classification loss $\mathcal{L}_{cls_{se}}$ and the RoI feature classification loss $\mathcal{L}_{cls_{roi}}$. The experiments in Table 4 shows the semantic anchor clustering loss and the semantic anchor classification loss are both indispensable in identifying unknown objects.

## 5 CONCLUSION

Open world object detection raises two challenging problems on current detectors, unknown recognition and incremental learning. Different from previous methods which combine domain-dependent technologies to respectively solve these two problems, our work provides a unified perspective, *i.e.* semantic topology. In the framework of semantic topology, all object instances from the same category, including 'unknown' category, are assigned to their corresponding pre-defined centroids. Therefore, a discriminative and consistent feature representations are ensured during the whole life of an open-world object detector. Aided with our proposed semantic topology, a huge improvement in open-world object detection performance across the board is achieved.

## 6 ACKNOWLEDGEMENT

Ping Luo was supported by the General Research Fund of HK No.27208720 and the HKU-TCL Joint Research Center for Artificial Intelligence.

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
