# OpenReview forum: "Objects in Semantic Topology"
_ICLR.cc/2022/Conference — ICLR 2022 Poster_

### Official Review · Reviewer_JDQB · 2021-11-02

**Correctness:** 3
**Technical Novelty And Significance:** 3
**Empirical Novelty And Significance:** 3
**Recommendation:** 8
**Confidence:** 4

**Main Review:**

### Strengths

- The paper deals with an interesting problem, which has clear practical relevance
- Overall, the approach is well described and motivated, with some details that need further improvement (see below)
- The proposed method seems technically sound
- The results show a clear improvement over prior work


### Weaknesses

#### Clarity

- Eq. 2 is not fully defined. What is the additional semantic feature classification loss exactly? What is its label space? Also the "traditional detection" losses need to be explained because this is a continual learning setup and not a "traditional" detection setup. Are boxes that RPN defines as "unknown" a separate class in the RoI classifier? Or are only the currently known categories used?

- Is there any further discussion about why three classifiers (traditional detection classifier, semantic feature classifier and the clustering itself is essentially also a classifier) are needed, besides the empirical ablation study? Some form of justification would be good.

- The last sentence of section 3.4.1 raises the question what the embedding is for the "unknown" class.


#### Related work

- How is the concept of "semantic anchors" different to approaches in zero-shot learning/detection that also encode the category names with a language model and use the embeddings as the classification/regression targets?  For example [A,B,C]

- Another aspect to consider in the related work are methods that try to combine multiple object detection datasets with different label spaces to create a larger unified label space, like [D,E].


#### Typos, minor comments and suggestions

- "has arisen" in abstract is maybe the wrong word? What about "has raised ... interest"?
- "open-set error" in abstract is unclear for the general reader I guess. I would express this in a different way or give a relative improvement.
- Typo: "making such detectors cannot handle"
- Is there a reference or a discussion to support the statement that the non-end-to-end ORE approach is "far from optimal"? This split into out-of-distribution detection and incremental learning seems natural.
- 3.2: I would write out "SA Head", I assume it means "Semantic Anchor Head".
- Why not use cosine similarity between RoI feature embedding and text embedding and do a cross-entropy loss over the similarities instead of Eq. 1? That's similar to how CLIP aligned images and text.
- 4.1 What does "semantic drifts" mean?
- 4.2 The recall level R is not described what it does
- 4.4 Typo: "at the following task"
- 4.5 Meaning of "group of classes (10, 5 and the last class)" is unclear
- 4.6 Typo: "To be specific, ..."


#### References mentioned above

- [A] Zero-shot object detection. Bansal et al. ECCV'18
- [B] Open-Vocabulary Object Detection Using Captions. Zareian et al. CVPR'21
- [C] Open-Vocabulary Object Detection via Vision and Language Knowledge Distillation. Gu et al. arXiv'21
- [D] Object Detection with a Unified Label Space from Multiple Datasets. Zhao et al. ECCV'20
- [E] Simple multi-dataset detection. Zhou et al. arXiv'21


**Summary Of The Paper:**

The paper proposes a novel method for open-world object detection, where instances of unknown categories need to be identified and annotated data for such new categories need to be integrated into the model in an incremental fashion. Prior work typically separate this problem into two tasks, out-of-distribution detection and incremental/continuous learning. This paper proposes to use fixed semantic anchors for each category, which are embedding vectors from a language model (or randomly generated vectors) for each category.  Importantly, when new data arrives, new embeddings are added while previous ones do not change. This encourages the feature representation to be compact (discriminative) and consistent (over time).

**Summary Of The Review:**

Overall, I think the paper presents a solid contribution to the task of open-world object detection. Some aspects of the paper need more clarification and some related work is missing that needs to be discussed.

---

> ### Author Response · Authors · 2021-11-11
> **Response to Reviewer JDQB**
>
> We thank the reviewer for the detailed reviews. We provide our responses below:
>
> ### Clarity
>
> **1. Regarding Eq. 2**
>
> The semantic feature classification loss is a cross-entropy loss built upon the semantic features with the same label space of the RoI feature classification head. Both the semantic feature and RoI feature classification heads define 'unknown' as one of the classification results. We have made them more clearer in the revised manuscript.
>
> **2. Why three classifiers?**
>
> Due to the dimensional discrepancy, we cannot directly manipulate the RoI features to construct the semantic topology by aligning with language models. Therefore, we used an additional fully connected layer to reduce the dimension of RoI features to the dimension of the language model, then the semantic anchors are explicitly used for constraining the semantic features with reduced dimension. The semantic feature clustering loss was used for constructing the semantic topology and the semantic classification loss was used to better leverage the learned semantic feature for classification. We have added more discussions in the revised paper.
>
> **3.What the embedding is for the "unknown" class.**
>
> As the same as all other known classes, we simply embed the 'unknown' word into the language model to generate its corresponding anchor.
>
> ### Related works
>
> **1.Compared with zero-shot learning**
>
> We have added extra related works on zero-shot learning as you suggested. Though our proposed method leverage a pretrained language model to generate the classification targets as many zero-shot learning methods did, we would like to highlight that our main idea is that **a consistent feature manifold topology plays an essential role in open-world object detection**, and the language model is just a perfect implementation of our idea. A consistent feature space topology helps stabilize previous-learned knowledge when learning new classes, which is highly required in open-world object detection. To verify this idea, we conduct experiments by randomly generating anchors to constrain the feature space learning in Table.3 and achieve surprising results. However, some fatal disadvantages make the randomly generated anchors not practical, (a) it cannot guarantee all anchors keep a decent distance so that the learning is effective, (b) the randomly assigned class centers would hinder the learning ability of networks. Instead, we found a pretrained language model is perfectly suitable for deriving consistent and meaningful feature space topology.
>
> Compared with zero-shot learning, in which language model is the key component to transfer knowledge, the language model in our proposal is just a simple yet effective implementation of our idea of consistent topology. Our paper takes advantage of language models that inherently have the ability to depict relationships among the infinite class in the world, to generate consistent and growth-able feature manifold topology for open-world object detection. We have added more discussions in the revised manuscript.
>
> ### Typos, minor comments and suggestions
>
> Thanks for your comments and we have carefully addressed all typos in the revised draft. We provide responses for your minor comments below:
>
> **1.Non-end-to-end ORE**
>
> The OOD detection in ORE requires to compute energy score by accessing all class data distribution (including classes from future tasks), which is not much applicable in practice. While our proposed method has a much simpler and end-to-end pipeline to perform unknown recognition.
>
> **2.Cosine similarity**
>
> Empirically, we found that directly feeding all semantic features into a classifier performs slightly better.
>
> **3.Semantic drifts**
>
> We follow the dataset split in ORE, which groups the 80 classes (20 from VOC and 60 from coco) into four tasks. We show the semantic split here:
>
> | Task1  | Task2  | Task3| Task4 |
> |  ----  | ----  |----  |----  |
> | VOC Classes | Outdoor, Accessories, Appliance, Truck | Sports, Food| Electronic,Indoor, Kitchen, Furniture|

---

> > ### Comment · Reviewer_JDQB · 2021-11-24
> > **Thanks and follow-up questions**
> >
> > Dear authors, thanks for the detailed response and the clarifications.
> >
> > Two follow-up questions:
> > - Thanks for the clarification about my question on the necessity of three classifiers. I understand why there is a dimensionality mismatch between ROI features and the text embeddings. But I still do not understand why the two classifiers with losses L_cls_ce and L_cls_roi are needed? They both have the same label space, right? So, in theory one of them would be sufficient, no? For instance, you could drop the classifier on the ROI features and your method would still work. I understand if having the two classification losses gives you better results. But if better results is the only reason to keep the two classifiers, this should be clarified in the paper.
> > - Regarding the "unknown" class: It's an interesting approach that you simply put the word "unknown" through the CLIP-text encoder and use that as your embedding. I'm wondering if that's the optimal choice. Prior works on zero-shot object detection learn the embedding for the ambiguous "background", i.e., initialize a free parameter (vector with dimension equal to the text embeddings) and learn it through standard back-propagation. Some even learn mixtures of background embeddings, like Bansal et al. Zero-shot object detection. ECCV'18

---

> > > ### Author Response · Authors · 2021-11-25
> > > **Response**
> > >
> > > Dear Reviewer,
> > >
> > > Thanks for your response and further comments!
> > >
> > > For the first question, yes, we ablate each classifier in Table.3 and empirically found combining them performs slightly better. By dropping the RoI classifier, our method still works significantly better than baselines. We would make it clear in the paper.
> > >
> > > Thanks for your great comments! It is very interesting that design the 'unknown' anchor in a learnable way or by mixing several embeddings, we will explore and discuss more about this.
> > >
> > > Thanks,
> > >
> > > Authors

---

> > > > ### Comment · Reviewer_JDQB · 2021-11-29
> > > > **Acknowledging author feedback**
> > > >
> > > > Dear authors,
> > > >
> > > > thanks for answering my follow-up questions. Regarding the design of the "unknown" anchor, I would suggest to add a small discussion into the paper and mention the related work.
> > > >
> > > > Thanks

---

> > > > > ### Author Response · Authors · 2021-11-30
> > > > > **Thanks!**
> > > > >
> > > > > Dear Reviewer,
> > > > >
> > > > > Thanks for your valuable comments! We will definitely discuss the choice of ‘unknown’ anchor in the revised paper.
> > > > >
> > > > > Best,
> > > > >
> > > > > Authors

---

### Official Review · Reviewer_8C8m · 2021-11-02

**Correctness:** 3
**Technical Novelty And Significance:** 2
**Empirical Novelty And Significance:** 2
**Recommendation:** 5
**Confidence:** 3

**Main Review:**

**Pros**

* The core idea introduced in this paper is nice, simple, and intuitive: simply leverage large-scale language models for zero-shot transfer to initialize a large variety of semantic class prototypes in the embedding space. Such augmentation of ORE makes perfect sense, and as verified experimentally, is very effective!
* This paper tackles and very challenging important problem and the methodology is presented clearly.
* I like the experimental evaluation; besides showing that leveraging language models for this task improves performance on incremental object detector learning, this paper also nicely ablates the impact on the effect of different language models.


**Cons**

* I am confused about the claims that prior work does not encourage intra-class compactness; isn't that exactly what the contrastive loss (in ORE) (in this paper referred to as clustering loss) is supposed to do?
* This paper adopts the task description from [A], claiming that novel objects are recognized as unknowns and should be labeled by annotators and used for re-training. This is not what the proposed method or method being built upon (ORE) does. Both are always only evaluated for the task of incremental learning, in terms of mAP and metrics, that evaluate confusion between known and unknown classes (e.g., Wilderness Impact measure): neither [A] or this paper provide any evidence that the learned detector is effective in detecting novel object instances (and separating them from the background *stuff* classes), and can be reliably be used in (for example) in conjunction with active learning to minimize the annotation effort. In summary, there is no evidence that any of the "new" classes in the new task sets are actually being detected before the network update.
To justify the claims made in Sec. 3.1, the model would need to label instances as unknowns and transfer semantic labels to these before using them for the model re-training (and not simply usefully labeled images for the new task set). I know that this confusion was not introduced in this paper. Rather, this paper just iterates claims and statements made in [A]. Still, I would suggest not propagating it further and fairly acknowledge that this paper really tackles object detection in an incremental learning setting.
* The basis of this paper is zero-shot learning: the knowledge about unseen classes is transferred from the language models. Such a paper absolutely must establish links with zero-shot learning and discuss prior work. I can understand that in the current rate of publications in our field one can miss a certain related paper, and I am usually happy to point this out and suggest updating the related work. However, this is not the case here; this paper entirely ignores the field of zero-shot learning, which is a basis for this work.
I strongly suggest performing a literature review in this field. In terms of zero-shot recognition, a good source would be [D]. Closely related methods on zero-shot detection and segmentation are aforementioned [C], and [E].
* A big part of the paper is 1-1 re-write of sections from [A], e.g., 3.1, 3.3, 4.1 (using own words). A fair thing to do would be to write a very short recap on problem definition and ORE detector, followed by introducing what is novel in this work. Reading this paper largely feels like re-reading [A].


**Question:** which semantic classes are used to initialize the anchors in the embedding space?


**Summary Of The Paper:**

This paper extends the recently-introduced object ORE object detector [A]. ORE is trained in an incremental fashion and was shown to minimize the confusion between classes presented in different task sets by adding a contrastive objective to the model training, that pushes features, representing different semantic classes, far away in the feature space. This paper presents the case that combining ORE detector with large-scale language models, trained by aligning textual queries and images, can significantly improve ORE detector results. This is not surprising: it was already shown that such multi-modal language models could be successfully used for detection of novel classes in a zero-shot setting (in which names of target classes are given, but image data for these classes is available during the model training, see [D]).
To the best of my knowledge, the core idea of semantic anchoring using language models was introduced in [C] (see Sec. 3.3 Densely Sampled Embedding Space). In this work, the semantic embedding space is extended with additional data from external sources that contain semantic information about the unseen classes via language embedding to image embedding alignment. This method suggests using CLIP [B] instead (which makes sense and simplifies the knowledge transfer!)

[A] Joseph et al., CVPR’21
[B] Radford et al., ICLR’21
[C] Bansal et al., ECCV’18
[D] Xian et al., CVPR'17
[E] Zheng et al., CVPR’21

**Summary Of The Review:**

This paper describes a nice way of leveraging knowledge about various visual concepts, distilled in modern language models, to improve over incremental learning method by [A].

However, overall I, unfortunately, need to point out that the proposed (valuable and effective!) extension of ORE is almost exactly what was proposed before in the scope of zero-shot object detection [C], just leveraging the most recent developments in natural language processing. In case my interpretation is wrong I would like to see a very thorough discussion in relation to the aforementioned (that should already have been in the original manuscript), with a focus on what exactly the difference is.


I still think this paper still carries an interesting message for the community, but (i) link to prior work on ZSL must be clearly established, and (ii) I do not think that ICLR is the right venue for this work in terms of novelty.


**Post rebuttal**


I would like to thank the authors for their comments and colleagues for the discussion. First, I would like to clarify that the paper was updated to include a discussion on the closely related topic of ZSL. I take back my comments that the usage for semantic anchors reduces this problem from open-world detection to zero-shot detection. Those were based on my initial miss-conception on how semantic anchors were used.

However, I still think that the evaluation protocol studies only incremental learning and provides no evidence whether the proposed method can be actually applied to the (significantly more challenging!) problem of open-world detection, and authors actually do acknowledge that in one of their responses.

As a reviewer, I cannot recommend accepting a paper that (in my view) has a mismatch between paper claims and premise (open world detection) and the actual delivery (incremental learning). My colleagues do not find this point as concerning (esp. given the fact that this issue was inherited from the prior work), therefore I will not strongly argue against acceptance, but I nevertheless wanted to bring this issue to AC's attention.

---

> ### Author Response · Authors · 2021-11-11
> **Response to Reviewer 8C8m**
>
> We thank the reviewer for the detailed reviews. We provide our responses below:
>
> **1.Intra-class compactness**
>
> In the paper, we claimed that the intra-class compactness and feature manifold topology consistent are two key components in open-world object detection. We use Fig.1 and Fig.2 to show the effect of compactness and consistency compared with vanilla Faster R-CNN. Yes, you are correct, the contrastive loss in ORE encourages intra-class compactness, we will re-discuss this in the revised manuscript.
>
> **2.The open-world object detection v.s. incremental object detection**
>
> We would like to kindly highlight the major difference between open-world object detection and class incremental object detection lies in the open-set recognition. In open-world object detection, the detector is **required to identify unknown objects that have never been learned by the detector and label them as 'unknown' as in Figure.4**. While in incremental detection, the detector can simply **ignore all unknown objects, in other words, categorize them as background**.  In Figure.4 (Ours T1), though the giraffe has never been involved into the detector, the detector can still recognize them and label them as 'unknown', while in T4 our detector can successfully label the objects as 'giraffe' since this class was introduced to the detector for re-training.
>
> To measure the ability of such unknown detection which differs from traditional incremental learning, the WI and A-OSE were introduced. The A-OSE counts the absolute number of unknown instances that are wrongly classified as any of the known classes, the WI measures the ratio of precision computed on known classes and the union of known and unknown classes, which evaluates the detector's ability to successfully detect unknown objects and classify them as ’unknown’ class. An incremental learning detector that cannot separate unknown objects from the background would result in high WI scores.
>
> **3.Compared with zero-shot learning**
>
> We appreciate your comments, and we have added extra discussions on zero-shot learning in the related work,  but **we respectfully disagree with you that the basis of our paper is zero-shot learning.** We would like to highlight our key viewpoint and contributions here, in this paper, we identify that **a consistent feature manifold topology plays an essential role in open-world object detection**.  Because a consistent feature space topology helps stabilize previous-learned knowledge when learning new classes. To verify this idea, we conduct experiments by randomly generating anchors to constrain the feature space learning in Table.3 and achieve surprising results. However, some fatal disadvantages make the randomly generated anchors not practical, (a) it cannot guarantee all anchors keep a decent distance so that the learning is effective, (b) the randomly assigned class centers would hinder the learning ability of networks. Instead, we found a pretrained language model is perfectly suitable for deriving consistent and meaningful feature space topology.
>
> Compared with zero-shot learning, in which language model is the key component to transfer knowledge, the language model in our proposal is just a simple yet effective implementation of our idea of consistent topology. Our paper takes advantage of language models that inherently have the ability to depict relationships among the infinite class in the world, to generate consistent and growth-able feature manifold topology for open-world object detection. We have added more discussions in the revised manuscript.
>
> **4.Re-write of sections**
>
> To make our paper easy-to-read and self-contained for those readers that have never studied ORE, we tried our best to describe the Section 3.1 problem definition, and Section 4.1 Datasets as detailed as possible. We will find a way to recap the related works and add more discussions.
>
> **5.Which semantic classes are used to initialize the anchors in the embedding space?**
>
> We initialize the semantic anchors by embedding the corresponding class names into a pretrain language model. The initialized anchors are kept unchanged during the whole life of open-world learning.

---

### Official Review · Reviewer_pp7T · 2021-11-07

**Correctness:** 3
**Technical Novelty And Significance:** 2
**Empirical Novelty And Significance:** 3
**Recommendation:** 5
**Confidence:** 3

**Main Review:**

- Marginal novelty.
While the semantic topology scheme is somewhat new, this work is heavily built on the work of Joseph et al., CVPR’21. Many parts of this paper are recaps of Joseph et al.'s work. Furthermore, leveraging the embeddings of the pretrained language model is popular in the zero-shot learning literature, so I don't think the core idea is very original while it's interesting to see the effect of the simple method.

- Predefined and rigid semantic topology.
The proposed semantic topology is rigid and not adaptive to the new object classes coming along. While it's surprising that it substantially improves the performance, this predefined and fixed topology is too simplistic and may not scale well to a larger number of classes.

- Use of a pretrained language model.
The proposed semantic topology requires the use of a pretrained language model, which means an external knowledge source as well as the use of class 'word' information in training. In this aspect, the performance comparison to the previous work can be seen as not fair. While the authors provide the result with random anchors for semantic topology, the results are not complete and also make me intrigued. Since the random anchor assignment may organize the embedding space in a random manner, it may hinder the learning of a proper embedding space. It needs more careful investigation on this issue.

- Incomplete analysis.
Some important studies on the effects of language models and semantic anchor losses are done only on task #2, which is strange. To see the consistency along with the incremental learning processes, the results need to be shown for all the tasks. In particular, the study on the random anchors needs more clarification and in-depth analyses. How exactly the random vectors are generated? If the random vectors are too close to each other, they may harm the performance, in particular, when the number of classes increases. Did the authors observe such effects? Did the author use a specific random assignment to make the anchors distant from each other?

- Missing details.
Unknown-aware RPN in Section 3.3 is not clearly described. Is it trained or just borrowed from the work of Joseph et al.'s 2021? Since this looks like one of the core modules in this work, a more detailed description needs to be done for the paper to be self-contained.

- Etc.
The title of this work is too general for readers to recognize what this work is about. I suggest changing the title and making it more informative, at least, including the term 'open-world detection'.


**Summary Of The Paper:**

This paper proposes a semantic topology embedding for Open-World Object Detection (OWOD) where an object detector identifies objects of unknown classes and incrementally learns to classify them assuming that their annotations are progressively given by humans. To maintain discriminative and consistent relationships among object classes, the authors introduce a semantic topology for the feature space of the detector by constructing pre-deﬁned anchors for categories using a pretrained language model. During training, it enables the detector to distinguish unknown objects out of the known categories and also makes learned features of different classes undistorted during incremental learning. Eperimental results show that the semantic topology improves state-of-the-art open-world object detectors and help the open-world detectors preserve a discriminative and consistent feature space.

**Summary Of The Review:**

While the proposed semantic topology is simple yet effective, outperforming the state of the art, this work has marginal novelty, clear limitations, and missing analyses. I'm on the borderline, slightly leaning toward rejection. I will make a final decision based on the authors' rebuttal responding to my critiques.

---

> ### Author Response · Authors · 2021-11-11
> **Response to Reviewer pp7T**
>
> We thank the reviewer for the detailed reviews. We provide our responses below:
>
> **1.Marginal novelty.**
>
> This paper follows the problem setting of open-world object detection, which was proposed in [A]. From the technical perspective, the only overlap between our proposed method and [A] is that we adopt an unknown-aware RPN from [A]. In this paper, we identified that **a consistent feature manifold topology is crucial for open-world learning.** By constraining the feature space to be consistent during the whole life of an open-world detector, the novel class learning would be more effective and won't damage the learned knowledge. To construct such a consistent feature space, we propose to leverage a pre-trained language model due to the following reason: a language model inherently has the ability to model the relationship among infinite object classes in the world, making it perfectly suitable for open-world feature space construction.
>
> Compared with [A], which accesses data distribution of all classes (including future classes) to identify unknown in a two-stage process, our proposed method significantly simplifies the unknown detection by constructing a consistent feature space topology and improves the performance by several times.
>
> Compared with zero-shot learning, leveraging a language model is not our key contribution, it is just a specific implementation of our proposed consistent feature space topology, we also verified the effectiveness of the idea of consistency by randomly constructing the feature space in Table.3.
>
> [A] Towards Open World Object Detection, Joseph et al., CVPR 2021
>
> **2.Predefined and rigid semantic topology.**
>
> Sorry, but we would like to argue that our proposed semantic topology is **growth-able in the number of classes** but **consistent in anchor locations**, which means the number of nodes (semantic anchors) in the topology would be increased during new class learning while the locations of all old anchors keep consistent during the whole life of the detector. To satisfy the requirements of growth-able and consistent, the semantic topology is derived from a pre-trained language model, which inherently has the ability to depict the relationships among infinite object classes around the world.
>
> **3.Use of a pretrained language model.**
>
> We respectfully argue that the use of pretrained language model is acceptable here. The reason is that,
> (a) The pretrained language model provide perfect priors for the challenging open-world learning, the ability of modeling 'the whole world' makes the language model perfectly suitable for open-world vision learning. (b) It raises no extra cost except a few lines of code to extract information from a pretrained language model. (c) Introducing semantic priors (e.g. WordNet or language models) to boost challenging vision learning is very common and effective, e.g. zero-shot image classification or fine-grained image classification.
> We expect this simple but effective implementation could serve as a solid baseline for future work in open-world object detection.
>
> We totally agree with you that 'The random anchors may hinder the learning of a proper embedding space.' Actually, the random anchor experiment was only used to verify our idea of 'A consistent feature manifold topology is a key role in open-world object detection.' The experiments verified the idea holds. However, some fatal disadvantages make the randomly generated anchors not practical: (a) it cannot guarantee all anchors keep a decent distance so that the learning is effective, (b) the randomly assigned class centers would hinder the learning ability of networks. Instead, a pretrained language model can perfectly overcome these problems.
>
> **4.Incomplete analysis.**
>
> Thanks for your suggestion and we have included extra experiments on all tasks in the revised manuscript. Please see Table.3 and Table.4. Regarding the random anchors, to keep them at a certain distance to avoid overlap, we randomly sampled all anchors from the 512-dimensional feature space uniformly, that is to say, we assume the total number of classes is known which is not applicable in practice. Therefore, we leverage language models to generate semantic anchors which is flexible on class numbers and make more sense.
>
> **5.Missing details.**
>
> We borrowed the unknown-aware RPN from [A] and trained the whole network end-to-end with our proposed semantic topology. We will discuss more this to make our paper self-contained.
>
> **6.The title.**
>
> Really thanks for your valuable comments and we have changed our title to 'Semantic Topology for Open-world Object Detection'.

---

### Official Review · Reviewer_fqMP · 2021-11-08

**Correctness:** 4
**Technical Novelty And Significance:** 2
**Empirical Novelty And Significance:** 3
**Recommendation:** 8
**Confidence:** 4

**Main Review:**

#### Strengths:
1. The idea to use semantic topology to enforce discriminative features that are consistent across incremental learning is an interesting direction for open set detection. The paper is well written and the ideas and experiments are presented with clarity.
2. Experiments are thorough, including ablations that show the effectiveness of different model components. It is interesting to see that using an additional classification loss on a separate stream of clustered RoI object features leads to significant improvement in detection performance.

#### Weaknesses:
1. The proposed approach seems to show significant improvement in detecting unknown objects. However I have a few questions on Table 1 results:

     (a) Wilderness impact values of ORE [A] in Table 1 is ~2 times that of the corresponding values in the [A], while all other results seem comparable. What is the reason for this?

     (b) Task 2 Wilderness index for ORE [A] is shown as 0.2970. (Should it be 0.0297?) The number seems inconsistent, as it is an order of magnitude higher than other comparable results in the table(for task 1 and task3), and also the corresponding result from [A]. The same result is also shown in Table 3.

      (c) If (a) and (b) do point to some errors, it seems that there are significant gains in A-OSE but much less gain in WI when comparing with [A]. Could this be explained based on the data?

2. The proposed approach is closely related to [A], which is also the main baseline. However, Fig 1 and Fig 2 show the learned representations using vanilla training strategies using Faster R-CNN vs. using semantic topology. Compactness of feature representation is also obtained in [A], so the figure tends to mislead the reader about prior work. It would be better to see a comparison of feature representations of semantic topology vs. that of ORE.

#### Additional comments / Questions:

1. Does semantic feature classification head also predict for the “unknown” class (unlike RoI feature classification)? It seems to be implied that way, but it is not made explicit in the paper. It would be useful to have a description on exactly how “unknown” objects are detected during inference.

2. Formula of WI in subsection 4.2 is not correct (-1 should be a separate term, not in the denominator). Also, the citations for WI and A-OSE are swapped in that section.

3. The model is designed such that features of unknown objects will change during incremental learning to cluster around predefined semantic anchors. It would be good to have more discussion on the nature and choice of unknown anchor in the semantic topology. For instance, what is a good choice for the unknown anchor? Does the choice of unknown anchor affect learning?

[A] Towards Open World Object Detection, Joseph et al., CVPR 2021


**Summary Of The Paper:**

The paper addresses the problem of open world object detection - a lifelong learning system where the object detector is required to detect objects belonging to all classes known so far and “unknown” objects. Unknown objects are annotated and available for training at a later stage. At a given time only a specific set of training data is available with annotations that involve all known classes so far, while all past training data are unavailable.


The proposed approach builds upon recent work [A], by incorporating a semantic topology in the object feature space (RoI features in Faster R-CNN). While [A] use contrastive clustering to group object features based on object category, and energy based out-of-distribution detection to detect “unknown” objects, the authors propose an end-to-end trainable approach. The proposed approach uses RoI features in two parallel streams, with one focused on clustering operation, while having object category prediction loss on both streams of features. Features are clustered around predetermined semantically meaningful object representations. Pre-trained word embeddings from a language model (CLIP) are used for this purpose. The use of pre-defined semantic anchors allows learned features of known objects to form consistent clusters during incremental learning.


Experiments are conducted using 80 categories from PASCAL-VOC and MSCOCO datasets. The problem is designed as 4 sequential tasks each involving 20 object categories (same setup as in [A]). Detection performance is measured using mAP, as well as measures that focus on unknown object detection.

[A] Towards Open World Object Detection, Joseph et al., CVPR 2021



**Summary Of The Review:**

The proposed approach is significant and relevant in open world object detection, proposing an end-to-end approach that learns incrementally and can detect unknown objects. A key promise of the proposed approach over prior work seems to be the improved ability in detecting unknown objects. However, I have concern regarding the main results reported (see weakness 1), and some other questions. I am willing to raise my rating if the authors can address these.

**[Edit after discussion period: ]** I thank the authors and the other reviewers for active engagement in the discussion. Important related work on zero shot learning was not cited in the preliminary version, which the authors included later on. While there is one point of concern in the paper claims (see below), I think this work provided a solid contribution to open-world detection problem. Hence I would like to raise my rating assuming the authors can make the necessary updates to fix the concern below.

* It was brought to notice by a reviewer the discrepancy between the problem definition in Section 3.1 and the evaluation protocol. While the problem is defined to involve "human user.. annotate (unknown instances) and return back to the model" at each stage, the evaluation protocol involves a different training set involving the unknown instances at each stage. I believe both settings are equally relevant for open-world detection problem, and I would encourage the authors to update the claims to match with the actual evaluation protocol.

---

> ### Author Response · Authors · 2021-11-11
> **Response to Reviewer fqMP**
>
> We thank the reviewer for the detailed reviews. We provide our responses below
>
> ### Weaknesses:
>
> **1.(a)&(b) The inconsistent results**
>
> Yes, it is a typo, the WI of ORE in Task 2 should be 0.0297, we will correct it in Table 1 and Table 4.
>
> We have tried our best to align our reproduced results with those reported in [A]. We re-run all experiments using the official released code and dataset to make a fair comparison. The reason why our reproduced results are different from ORE [1] is the different data split (not the class split). We have connected with the authors of ORE, the authors admitted that the released data split is different from those they used in the paper, even worse, they lose the data split used in the paper. We also find other people meet a similar reproducing problem in the issues of ORE GitHub repo (https://github.com/JosephKJ/OWOD/issues/26).
>
> **1.(c) Less gain in WI**
>
> A-OSE counts the absolute number of mislabeled unknown instances, while WI measures the ratio of precisions. Though our proposed method can simultaneously improve these two metrics, but their improvements maybe not in proportional.
>
> **2.Fig.1 and Fig.2**
>
> In Fig.1 and Fig.2, we compared our method with vanilla training strategy to **explain the concept** of feature manifold topology consistency and feature representation compactness. We sincerely admit that the feature representation compactness was also used in [A] and we didn't claim it as one of our main contributions. We are sorry if we misled readers about prior works and we have re-organized the claimed contributions in the revised paper.
>
> ### Additional Comments:
>
> **1. How “unknown” objects are detected during inference?**
>
> Both classification heads built on top of RoI features and semantic features were designed to have C+1 outputs for predicting C known classes and one unknown class.  At the inference stage, the classification results are computed by multiplying the two class posterior probability vectors predicted by the RoI feature classification head and the semantic feature classification head.
>
> **2. The formula of WI?**
>
> Sorry for the mistake, we have corrected the formulation and the citation order.
>
> **3. The choice of unknown anchor**
>
> In our implementation, we simply embed the word 'unknown' into a pretrained language model to generate anchor for 'unknown' objects. This has several intuitions, (a) we generate semantic anchors for all classes by embedding their class names using a language model, the same strategy should be used for 'unknown' anchor to construct a consistent and meaningful feature space topology, (b) the word embedding of 'unknown' is relatively far from those of known classes, by clustering unknown objects around the 'unknown' word embedding, the feature space becomes more discriminative between known and unknown categories, thus improves the unknown detection ability. In Table.3, we conduct experiments by randomly generating all anchors including 'unknown' anchors, randomly generated anchors perform worse than semantic-guided anchors.

---

> > ### Comment · Reviewer_fqMP · 2021-11-19
> > **Fig 1 & 2 still misleading**
> >
> > I thank the authors for the detailed response. It has addressed most of my concerns.
> >
> > However maintaining Fig. 1 & 2 the way they are is misleading, as it implies that the idea of having feature compactness is a contribution of this work. It would be good to highlight in figures what is novel from prior work (Joseph et al.).

---

> > > ### Author Response · Authors · 2021-11-19
> > > **We have changed the Fig.1**
> > >
> > > Dear reviewer,
> > >
> > > We have revised the Figure.1 to the visualization of ORE according to your valuable suggestion. As the Figure.1 shown, ORE doesn't guarantee a consistent feature space topology during incremental learning. We have re-discussed the differences between our method and ORE in the introduction section.
> > >
> > > Since all your concerns have been addressed, would you like to kindly raise the rating?
> > >
> > > Thanks,
> > > Authors

---

> ### Author Response · Authors · 2021-11-30
> **Thanks!**
>
> Dear Reviewer,
>
> Thanks! We will fully discuss the problem setting and the evaluation protocol in the revised version.
>
> Best,
> Authors

---

### Author Response · Authors · 2021-11-11
**Summary of Revisions**

We really appreciate all four reviewers for their valuable comments.
We’ve uploaded a revised draft incorporating reviewer feedback. Below is a summary of the main changes:

1.We have changed the paper title to 'Semantic Topology for Open-world Object Detection'.

2.We have added discussions on zero-shot learning and claimed the core differences in Related Works (Section 2).

3.We have added extra ablation experiments on all tasks in Table 3 and Table 4.

4.We have addressed all clarity issues mentioned by four reviewers and polished our paper.


We really hope our responses and revisions address all reviewers’ concerns!

---

### Decision · Program_Chairs · 2022-01-20

**Decision:**

Accept (Poster)

**Comment:**

This paper presents work on open-world object detection.  The main idea is to use fixed per-category semantic anchors.  These can be incrementally added to when new data appear.  The reviewers engaged in significant discussion around the paper with many iterations of improvements to the paper.  Initial concerns regarding zero-shot learning were addressed, as were remarks on presentation and claims.

In the end the reviewers were split on this paper.  I recommend to accept the paper on the basis of the semantic topology ideas and the thorough experimental results.

The remaining concern centered around the evaluation protocol used in the paper, which follows that in the literature (e.g. Joseph et al. CVPR 21).  While this is not a fatal flaw, it is an issue with how this genre of methods is evaluated.  It would be good to add discussion to the final paper to highlight this as an opportunity for future work in the field to address.  Specifically, as a reviewer noted "after detecting "unknown" objects in T1, the (hypothetical) annotation process provides boxes for ALL objects of some new classes instead of only for those that have been correctly detected (localized and marked "unknown")."